

# Attribution of Greenland's ablating ice surfaces on ice sheet albedo using unmanned aerial systems

Jonathan C. Ryan[1], Alun Hubbard[1,2], Marek Stibal[3,4], Jason E. Box[5], and the Dark Snow Project team[*]

[1]Centre for Glaciology, Institute of Geography and Earth Sciences, Aberystwyth University, Aberystwyth, SY23 3DB, UK
[2]Centre for Arctic Gas Hydrate, Environment and Climate, Department of Geology, University of Tromsø, Dramsveien 201, 9037, Norway
[3]Department of Ecology, Faculty of Science, Charles University, Vinicna 7, 128 44 Prague, Czech Republic
[4]Department of Geochemistry, Geological Survey of Denmark and Greenland, Copenhagen, Denmark
[5]Department of Glaciology and Climate, Geological Survey of Denmark and Greenland, Copenhagen, Denmark
[*]A full list of authors and affiliations appears at the end of the paper

*Correspondence to:* Jonathan C. Ryan (jor44@aber.ac.uk)

**Abstract.** Surface albedo, a primary control on the amount of energy available for melt, has considerable spatial heterogeneity across the Greenland ice sheet ablation area. However, the relative importance of distinct surface types on albedo remains unclear. In this study, the causes of mesoscale ($10^2$ to $10^3$ m) albedo variability are assessed using high resolution (decimetre-scale) digital imagery and broadband albedo data acquired by a fixed-wing unmanned aerial system. We characterize the

reflectance properties and terrain roughness associated with six distinct surface types identified from a 25 km longitudinal transect across the ablating dark region of the Kangerlussuaq sector. Principal component analysis of the fractional area of each surface type versus coincident Moderate Resolution Imaging Spectroradiometer (MODIS) albedo data reveals the relative importance of each surface type. The highest correlation with mesoscale albedo was the fractional area of distributed impurities. Although not the darkest surface type, their extensive coverage meant that they could explain 65% of the albedo variability

across the survey transect including the presence of the dark region. In contrast, the 2% mean surface water coverage across our survey transect could only explain 12% of albedo variation and crevasses, only 17%. Localised cryoconite patches have the lowest albedo signature but comprise less than 1% of the survey area and do not appear to reduce mesoscale albedo. We anticipate further reduction in ablation area albedo under future warming as localized areas of distributed impurities, supraglacial water and crevassing increase in extent and conclude that current bare ice area albedo models may advance

significantly by representing the evolution of the surface types identified in this study.

## 1 Introduction

Bihemispherical albedo determines the amount of solar radiation absorbed at the surface and thus modulates the energy available for ice melt (van den Broeke et al., 2008). Spatial patterns of albedo are hence a key control on ice sheet mass balance and runoff contribution to global sea level. Albedo across the ablation area of the Greenland ice sheet is characterized by

high spatial heterogeneity, once bare ice is exposed, due to the influence of numerous non-ice constituents and surface structures (Knap and Oerlemans, 1996; Stroeve et al., 2006; Moustafa et al., 2015). These range from structural features such as





crevasses, fractures and foliation; supraglacial hydrology: rivers, streams, ponds and lakes (Greuell, 2000; Smith et al., 2015); snow patches and fracture cornices; cryoconite that is concentrated in either holes or in fluvial deposits (Bøggild et al., 2010; Chandler et al., 2015); microbes and their humic by-products (Takeuchi, 2002; Stibal et al., 2010; Yallop et al., 2012), mineral dust and aerosols from outcropping or contemporary aeolian deposition including black carbon from wildfires (Wientjes and

Oerlemans, 2010; Wientjes et al., 2011, 2012; Dumont et al., 2014; Keegan et al., 2014) and other aerosols. While the highest resolution optical satellite imagery currently available (WorldView-1 and 2 with a pixel footprint of 46 cm) have facilitated the examination of crevasse fields (e.g., Colgan et al., 2011) and surface hydrology (Smith et al., 2015), an assessment of how these constituent ice surface types aggregate to determine mesoscale ($10^2$ to $10^3$ m) albedo is yet to be made.

Remote sensing from unmanned aerial systems (UASs) bridges the scale gap between in situ point measurements and coarser

resolution satellite imagery. UASs can acquire high-resolution georeferenced imagery over tens of kilometres squared on daily or sub-daily intervals (Carrivick et al., 2013; Whitehead et al., 2014; Ryan et al., 2015; Bhardwaj et al., 2016). For example, Ryan et al. (2015) generate orthomosaics and digital elevation models at ground sampling resolutions of 12 cm and 50 cm, respectively, from repeat UAS surveys to investigate calving and flow dynamics across the terminus of Store Glacier, West Greenland. Likewise, Rippin et al. (2015) derived a 5 cm orthomosaic and 10 cm DEM across the lower reaches of Midtre

Lovénbreen , Svalbard to assess the relationship between surface roughness, surface hydrology and reflectance.

This study aims to relate the fractional area and albedo of ablating ice surface types to mesoscale ($10^2$ to $10^3$ m) albedo patterns in the ablation area of the Greenland ice sheet. To do this, we deploy a fixed-wing UAS equipped with a digital camera and pyranometers to survey a 25 km, flow parallel (longitudinal) transect over the Kangerlussuaq sector three times in August 2014 (Fig. 1). We then characterize the reflectance properties and terrain roughness of six distinct surface types and determine

their relative importance to coincident Moderate Resolution Imaging Spectroradiometer (MODIS) albedo data (Fig. 2).

## 2 Data and Methods

### 2.1 UAS platform and measurements

The fixed-wing UAS deployed is a modified version of the X8 airframe described by Ryan et al. (2015). It has a 210 cm wingspan and is powered by a 10 Ah, 16.8 V LiPo battery pack which, with an all-up take off weight of 4 kg, gives a 1

hour endurance and 60 km range. The autonomous control system is based around an Arduino navigation and flight computer updated in real-time by a 10 Hz data stream comprising of a GPS, magnetometer, barometer and accelerometer. These data are logged along with a timestamp for each activation of the digital camera shutter which automatically triggers when a horizontal displacement threshold is exceeded. The UAS was hand launched from an on-ice base camp at 67.08°N, 49.40°W, located at the site of the Institute for Marine and Atmospheric Research (IMAU), University of Utrecht S6 automatic weather station

(AWS). On return from each sortie, it was manually landed into a 10 x 5 m net. The UAS was preprogrammed to carry out a 25 km survey collocated over the K-(Kangerlussuaq) transect of the western Greenland ice sheet on 6, 7 and 8 August, 2014 (Fig. 1). The Greenland Ice Mapping Project (GIMP) DEM (Howat et al., 2014) was used during the selection of 3D waypoints to ensure the UAS flew at a consistent altitude of 350 m above the surface on the fully autonomous sorties.



## 2.2 Apogee SP-110 pyranometer albedo

Upward and downward facing Apogee SP-110 pyranometers were mounted into the airframe to record broadband albedo along the transect and calibrate the digital imagery (Sect 2.3). The pyranometers were levelled to ensure accurate measurement of downward and upward irradiance in level flight and were sampled at 1 Hz by a Campbell CR200x data logger. Variations in

UAS attitude, due to winds or course changes, that were deemed to adversely affect the measurements were filtered out of the observations when pitch and/or roll exceeded $6°$. Downward irradiance (including direct beam radiance) were subsequently calibrated by applying a data cluster normalization to compensate for the effect of moderate pitch and roll adjustments and change in solar zenith angle. Surface broadband albedo ($\alpha_{pyra}$) was determined over the 25 km transect by dividing the upward by downward solar irradiance values. Albedo derived from the outgoing and return legs were compared for consistency and

quality assessment.

The SP-110 pyranometer has a stated accuracy of 5% (http://apogeeinstruments.co.uk/), including a directional or cosine response error of 2% at $45°$ solar zenith angle. Therefore the surveys were conducted $\pm$ 1 hour of the local solar noon at ~13:30 Western Greenland Summer Time (WGST) to minimize directional error. The SP-110 incorporate a silicon photodiode and benefit from low weight (60 g per sensor) and response time of under 1 ms. However, the pyranometers have a relatively

narrow spectral sensitivity of 350 and 1100 nm and thereby capture just ~80% of the shortwave spectrum. Reduced spectral response potentially yields a significant error when determining the albedo of surfaces that do not have a flat response and reflect (or absorb) outside the instrument's range.

Broadband albedo derived from a pair of Kipp & Zonen CM3 optical pyranometers were compared with the SP-110 pyranometers to assess these errors and uncertainties. Kipp & Zonen CM3 pyranometers are based on carbon black thermopiles

that yield a flat spectral response between 300 to 2800 nm. A variety of surfaces: ice, water and cryoconite, were simultaneously measured by both Apogee SP-110 and Kipp & Zonen CM3 pyranometers and the resulting data were compared. A root mean square difference (RMSD) of 3.6% between the two instruments indicated a collinear response with no detectable bias. The CM3 pyranometer has a specified accuracy of 2% and hence, we determine that downward Apogee SP-110 pyranometer mounted on the UAS has an accuracy of 4.1% ($\sqrt{3.6^2 + 2.0^2}$) when measuring upward shortwave radiation from the ablation

area. The upward pyranometer measures downward shortwave radiation to an accuracy of 5%, therefore the absolute accuracy of $\alpha_{pyra}$ is 6.5% ($\sqrt{4.1^2 + 5.0^2}$).

## 2.3 Digital imagery

Digital imagery was acquired by a Sony NEX-5N digital camera vertically mounted inside the front of the airframe. The camera has a 16 mm fixed focus lens (53.1 by $73.7°$ field of view) which yields a ground footprint of approximately 525 x

350 m at an altitude of 350 m above the ground. The camera was set for a fixed 1/1000 shutter speed, ISO 100 and F-stop of 8 and triggered every 35 m to provide a ~90% forward image overlap. The relatively fast shutter speed minimizes image blur whilst the low ISO and F-number ensures maximum image quality in which even the brightest surfaces cannot saturate the image. These camera settings were kept constant for all surveys to facilitate image comparison and calibration. For each of the





three sorties flown, ~2000 RAW images were acquired at the camera's maximum (4912 x 3264) resolution which equates to a ground sampling distance of ~11 cm, once the images were corrected for barrel distortion (Sect 2.4).

## 2.4   Digital camera albedo

The albedo of surface types that characterize the survey transect was estimated from digital image pixels under the assumption that the RGB digital numbers (DNs) in the acquired RAW imagery are proportional to the number of photons intercepted by the Sony NEX-5N's 16.1 MP CMOS sensor (Lebourgeois et al., 2008). During processing, the absolute RAW DNs were preserved by converting each proprietary Sony RAW image to a 16-bit TIFF image using *dcraw* (http://cybercom.net/~dcoffin/dcraw/). A vignette correction mask was universally applied to compensate for image and lens distortion due to edge effects which were as high as 17.6% at the corners of some images. The correction mask was calculated from the mean vignette of all images acquired on the 6, 7 and 8 August. Barrel, or geometric, distortion was corrected using ImageMagick (http://www.imagemagick.org/) which utilised the coefficients stored in the EXIF data.

We corrected the images for changing illumination conditions during and between successive surveys by normalizing each image with the downward irradiance measured by the upward facing pyranometer. To do this, images of a 25 x 25 cm Spectralon white reference target were acquired every 10 minutes during UAS surveying. The relationship between the DNs of the white reference target and the downward irradiance recorded by the UAS were used to produce a calibration curve which was best fitted using a linear least squares regression ($R^2$ 0.85). The ratio of the reflected radiation recorded by the camera and the downward radiation estimated from the calibration curve allowed us to calculate the 'illumination-corrected' reflectance of each pixel. An empirically-derived albedo product, $\alpha_{camera}$, was then produced by fitting another linear least squares regression between the reflectance of surfaces within the illumination-corrected images and albedo of surfaces measured using the CM3 pyranometer from the ground. The slope (1.2) and intercept (-0.01) of the regression analysis was used to convert the illumination-corrected images to $\alpha_{camera}$. The two products were highly correlated with an $R^2$ of 0.94 but had an RMSD of 7.8%.

The uncertainty of $\alpha_{camera}$ is probably due to the fact that the camera's CMOS sensor only captures a visible band of shortwave radiation between ~ 400 and 700 nm whereas albedo by definition represents a broader spectrum between 280 and 4000 nm that comprises the full wavelength range relevant to the surface energy budget and melt. Corripio (2004) argue that ultraviolet (280 to 400 nm) and infrared (700 to 4000 nm) radiation make up a minor proportion of total shortwave energy (10 and 25%, respectively) and that actual albedo can be estimated from visible band imagery alone with under 9% uncertainty.

## 2.5   Orthomosaic and digital elevation model (DEM) generation

The illumination-corrected images were used to produce orthomosaics and DEMs for each survey using Agisoft PhotoScan Pro (Agisoft LLC, 2013) following the processing sequence described by Ryan et al. (2015). The images were georeferenced by providing latitude, longitude, altitude and attitude data recorded by the flight controller. The orthomosaics were resampled to a common ground resolution of 15 cm, and the DEMs were resampled to 50 cm resolution.





## 2.6 Surface classification

Visual inspection of the UAS imagery revealed that six surface types characterize the survey transect: i) clean ice, ii) uniformly distributed impurities iii) "deep" water, iv) "shallow" water, v) concentrated cryoconite, and vi) crevasses/fractures (Table 1). These surfaces were manually digitized in the transect orthomosaics based on RGB brightness and a binary layer that specified

whether or not the pixel was situated in a crevasse or fracture. Crevasse identification was facilitated through determination of the residual between the original 50 cm DEM subtracted from a 30 m Gaussian smoothed DEM. Negative anomalies with a vertical displacement greater than 1 m were identified as crevasses. Small cracks and fractures were detected on the basis on sharp RGB contrast, and were discriminated using an edge detector algorithm (Canny, 1986). Pixels within 2 m of a linear feature were identified as fractures.

The digitized features were used to train a supervised k-nearest neighbours (k-NN) algorithm from the *scikit-learn* Python module (Pedregosa et al., 2011) which was applied to the orthomosaics produced on the 6, 7 and 8 August 2014. All pixels were classified into one of the six classes using a majority vote based on the Euclidean distance to five equally weighted nearest neighbours. The efficacy of the k-NN classifier (95.7%) was evaluated by separating the data set into equal areas of training and testing samples, and subsequently determining the number of independent test samples that were correctly classified. The

mean fractional area of each surface type was determined by summing the pixels in each class and dividing by the total number of pixels in the orthomosaic. The transect orthomosaic was also divided into 50, 500 $m^2$ segments to assess how the fractional area of each surface type changed through space. The segments allow us to attribute the fractional areas of each surface type to each MODIS pixel along the transect. Finally, the results of the classification procedure were visually inspected to provide an independent assessment of quality and to ensure consistency throughout the dataset.

## 2.7 MODIS albedo

Mesoscale ($10^2$ to $10^3$ m) albedo patterns were determined from the daily MOD10A1 V005 albedo product, $\alpha_{\mathrm{MODIS}}$, available from the National Snow and Ice Data Center (NSIDC) (Hall et al., 2006). MOD10A1 is gridded in a sinusoidal map projection and has a resolution of 500 x 500 m. Coincidence between $\alpha_{\mathrm{MODIS}}$ and UAS data was established using bilinear interpolation of the surrounding four $\alpha_{\mathrm{MODIS}}$ values for every segment along the survey transect. We estimated that $\alpha_{\mathrm{MODIS}}$ have a RMSD of

7.0% based on a comparison between the GAP/PROMICE weather stations: KAN-L, KAN-M and KAN-U between 2009 to 2014 (van As et al., 2012). This is equivalent to that of Stroeve et al. (2006) and Wang et al. (2011) who estimated a RMSD of 6.7%. A net bias of -1.4% was measured over the entire 2014 melt season which provides some evidence of sensor degradation (Lyapustin et al., 2014; Polashenski et al., 2015) but does not affect the detection of relative mesoscale albedo patterns.





## 3   Results

### 3.1   Albedo products

Analysis of satellite and UAS albedo data show that $\alpha_{\text{MODIS}}$ represents the mesoscale albedo variability accurately enough for the purpose of this study (Fig. 3). For all three surveys, $\alpha_{\text{pyra}}$ and $\alpha_{\text{MODIS}}$ show good agreement with a mean RMSD of 3.4%

(Table 2). This is within the 7.0% uncertainty of $\alpha_{\text{MODIS}}$ calculated in Sect 2.7. We found no detectable bias between $\alpha_{\text{MODIS}}$ over the three survey days. This shows that the ice surface did not change significantly enough to measure and, as a result, the rest of the study focuses on the digital imagery and albedo data collected on the 8 August survey. The mean $\alpha_{\text{camera}}$ and standard deviation for the six surface types that characterized our survey transect are presented in Table 1.

### 3.2   Surface type variation along transect

The fractional area of the surface types vary across the transect (Fig. 6). Coverage by clean ice averages 51% in the lower, western half of the transect between 0 and 14 km. In the eastern half (14 to 25 km), clean ice coverage is almost half this at 27%, coincident with, what has been termed, the dark region (Shimada et al., 2016). Uniformly distributed impurities vary inversely to clean ice, with a higher fractional area in the eastern half (70%) than the lower, western half of the transect (56%). The fractional area of surface water (both deep and shallow) are well correlated ($R^2$ 0.94). At 23 km along the transect, a

supraglacial lake with an area of 0.83 km$^2$ covers 32% of the corresponding MODIS pixel. Elsewhere, at 19 km along the transect, a braided meltwater stream has a fractional area of 3.8%, but otherwise, surface water has a mean surface area of 2% across the transect. Crevasse density is highest at the lower, western end of the transect between 2 and 7 km and attains maximum coverage of 5% at 6 km along the transect. Eastward of 7 km there are crevasses and fractures are conspicuously absent according to our DEM and edge detector analysis. Absence of crevasses and fractures was confirmed by visual inspection

of this area in the orthomosaics. Concentrated cryoconite has a maximum aerial coverage of 1.6% at 5 km along the transect and a mean footprint of 0.6% along the entire transect.

### 3.3   Principal component analysis (PCA)

$\alpha_{\text{MODIS}}$ exhibits high spatial variability across the transect from 0.28 to 0.47 and appears to be related to the fractional area of certain surface types (Fig. 3). Therefore PCA was used to quantify the correlation between the surface types and spatial

albedo patterns of albedo, as represented by $\alpha_{\text{MODIS}}$. The first principal component (PC1) explains 65% of the variance in the $\alpha_{\text{MODIS}}$ signal and is strongly (r > 0.8) correlated with both the fraction the surface covered by uniformly distributed impurities and, as would be expected, is negatively correlated with the occurrence of clean ice. The second principal component (PC2), crevassing, accounts for 17% of the variance and the presence of surface water (PC3) explains 12% of $\alpha_{\text{MODIS}}$ variation.



## 4 Discussion

### 4.1 The dark region

A conspicuous feature of the survey transect is the low albedo ($\alpha_{\text{MODIS}}$ and $\alpha_{\text{pyra}}$ of $\sim 0.29$) in the eastern half of the transect between 13 and 23 km (Fig. 3, 6). The area corresponds with the dark region and associated foliation bands in the Landsat 8 image (Fig. 1) (Wientjes and Oerlemans, 2010; Wientjes et al., 2012; Shimada et al., 2016). Between 70 and 80% the dark region is classified as uniformly distributed impurities, with the remaining 20 to 30% predominately clean ice (Fig. 4A, 6). It has been hypothesized that the concentration of surface meltwater and cryoconite in holes is the driver of reduced albedo across the dark region of the western ice sheet (Oerlemans and Vugts, 1993; Knap and Oerlemans, 1996; Greuell, 2000; Wientjes et al., 2011). However, we find that the dark region has a very low fractional area coverage of surface water (< 1.0%), concentrated cryoconite (< 0.5%) and crevasses (0.0%). Instead, uniformly distributed impurities over flat, smooth ice is the predominant surface type that characterizes the dark region and is the primary agent responsible for the low $\alpha_{\text{MODIS}}$ values (Fig. 4A).

Uniformly distributed impurities are attributed to aeolian dust deposited during the early Holocene and outcropping today (Wientjes and Oerlemans, 2010; Wientjes et al., 2011, 2012) and/or ice algal blooms and humic material (Takeuchi, 2002; Stibal et al., 2010; Yallop et al., 2012). Samples from the ice surface reveal that the dark region is abundant with green algae (*Cylindrocystis, Ancylonema and Mesotaenium*) (Yallop et al., 2012) and imagery obtained by Yallop et al. (2012) indicate that the ice algal blooms have similar RGB signatures to our UAS images (Fig. 4A). The algae are characterized by a grey/brown hue due to the brown-to-purple coloured pigments surrounding the algae chloroplasts which act as a screening mechanism to down-regulate photosynthesis when exposed to the high intensity visible and ultraviolet radiation (Remias et al., 2012a, b; Stibal et al., 2012). The presence of microbes is supported by the high concentrations of modern organic carbon near the ice sheet surface across the dark region (Wientjes et al., 2011, 2012).

The extent of the dark region increased by 12% between 2000 and 2014 in southwestern Greenland (Shimada et al., 2016). Air temperature in this region also increased 0.128 degrees per year in the same period (Shimada et al., 2016). If the surface of dark region is composed of algal communities then increased temperature and/or meltwater production will likely promote further colonization and growth of ice algae across the surface yielding an increase in pigmented biomass and albedo reduction in the ablation area (Yallop et al., 2012; Lutz et al., 2016). Currently, however, the physical processes that control ice algal extent and its relationship with albedo are not well known.

### 4.2 Supraglacial streams and lakes

Supraglacial water contained in both lakes and channels explains just 12% of $\alpha_{\text{MODIS}}$ variance. This is explained by the fact that surface water comprises just 2% of the total transect area (Fig. 6). Our results are consistent with Smith et al. (2015) who found that surface water had a fractional area of only 1.4% across the Kangerlussuaq sector. Previously, the ponding and accumulation of meltwater across the ablation area was cited as a primary reason for the low albedo of the ablation area (e.g., Oerlemans and Vugts, 1993; Knap and Oerlemans, 1996; Greuell, 2000). However, our results indicate that variations in



supraglacial lakes and stream density do not govern $\alpha_{\mathrm{MODIS}}$ patterns and that the preponderance of meltwater is a consequence rather than a cause of darkening across the ablation area (Wientjes and Oerlemans, 2010).

Locally, i.e. in one MODIS pixel, surface water can have a significant impact on albedo due to its high absorbance properties ($\alpha_{\mathrm{camera}}$ 0.19 to 0.26). For example, the MODIS pixel that contains a large braided meltwater channel is associated with an albedo of 0.28. This is ~ 0.05 lower than the surrounding MODIS pixels which have < 1% surface water but consist of ice surfaces with similar RGB brightness (Fig. 4B, 6). Likewise, MODIS pixels that contain supraglacial lakes are likely to be associated with low albedo. Despite this, the largest area of surface meltwater identified along the transect in this study was not associated with a reduction in $\alpha_{\mathrm{MODIS}}$. A supraglacial lake at 23 km along the transect consists of shallow water with a mean $\alpha_{\mathrm{MODIS}}$ of ~ 0.26 and deeper water with a mean $\alpha_{\mathrm{MODIS}}$ value of ~ 0.19, each of which comprise 12 and 20% of the corresponding MODIS pixel, respectively (Fig. 4C). But the lake is not associated with a change in $\alpha_{\mathrm{MODIS}}$ because the lake is relatively narrow (260 m width) and only covers around 32% of the MODIS pixel. Another 44% consists of clean ice faculae either side of the water body which have a higher mean $\alpha_{\mathrm{camera}}$ value of 0.55 and are associated with lake shorelines (Fig. 4C, 6). Therefore the low albedo of the lake is offset by the high albedo of the ice and there is no change in $\alpha_{\mathrm{MODIS}}$.

Despite surface water not necessarily being a primary driver for the depressed albedo across the dark region, a relatively small expansion in the spatial extent of surface water would have a disproportionate impact on albedo and associated surface melt rates. For example, ablation rates at the bottom of ponded supraglacial water are estimated to be double that of bare ice surfaces due to the effect of enhanced shortwave radiation absorption (Lüthje et al., 2006; Tedesco and Steiner, 2011). Future atmospheric warming has been shown to not only increase the density of ponded supraglacial water, but also increase their duration and extent (Leeson et al., 2015). During years with higher summer temperatures, such as 2007, 2010 and 2012, lakes formed earlier in the season and occupied an area up to 40% larger than cooler summers such as 2006 (Fitzpatrick et al., 2014). Analysis of satellite imagery has revealed new supraglacial lakes forming at higher elevations in the ice sheet interior over the past decade (Howat et al., 2013; Fitzpatrick et al., 2014). It follows that increased ponding of surface water in lakes will drive net albedo decrease across an expanding ablation area in the future.

### 4.3 Crevasses

Crevasses enhance shortwave radiation absorption; radiative transfer modelling indicates that double the downward energy is absorbed by a crevasse relative to a homogeneous, flat ice surface (Pfeffer and Bretherton, 1987). The PCA applied to our transect dataset indicates that crevasses explain some 17% of albedo variability as represented by $\alpha_{\mathrm{MODIS}}$. The impact on albedo is well illustrated at 4 km from the western end of the transect where a transition into a crevasse zone yields a ~ 0.10 reduction in $\alpha_{\mathrm{MODIS}}$ compared to the adjacent flat ice surface (Fig. 5A, B). The size and density of crevasses determines the amount of radiation absorbed. One of the most densely crevassed areas across the study transect (Fig. 5A) has mean crevasse widths of 10 m with depths to 8 m as detected by our residual DEM analysis driving a localized albedo reduction ~0.10. Elsewhere, crevasses along the transect are smaller with a width of 5 m and depths of 3 m, and have a lower impact on $\alpha_{\mathrm{MODIS}}$ reducing it by up to 0.05. Radiative transfer modelling predicts that crevasses reduce albedo by between 0.10 and 0.25 (Pfeffer and Bretherton, 1987). This result is in accordance with the upper bound for albedo reduction in our study but it should be noted



that modelled crevasses are embedded in, and compared, against a flat, clean ice surface which is not the case in this study. Cathles et al. (2011) modelled crevasses with width to depth ratios similar to that of crevasses observed in this study and found that crevasses of this geometry have a melt enhancement factor of 1.14 to 1.20 at solar zenith angles of 45 degrees.

Tensile extension caused by ice flow acceleration or sharp gradients in basal or surface topography are a primary determinant of crevasse occurrence (Hubbard et al., 1998; van der Veen, 1999; Colgan et al., 2011). Observed patterns of ice flow acceleration are complex indicating high levels of spatial heterogeneity in ice sheet dynamic response to recent atmospheric forcing. For example, van de Wal et al. (2008) report a decadal record of net reduced ice flow as measured by a permanent network of GPS across the marginal 35 km of the K-transect. Whilst Doyle et al. (2014) report net ice sheet flow increase above the equilibrium line and up to 140 km into the interior in the same sector. It should be noted that fractures and crevasses form under transient, short-lived, tensile conditions that do not necessarily reflect long-term average flow conditions. Hence, the short-lived by intense summer melt-forced episodes of enhanced flow observed across the ablating margin have the potential to initiate new fractures well above the equilibrium line through longitudinal stress coupling. New fractures observed over the ice sheet and as high as Raven Camp, located at 1800 m.a.s.l and ~170 km from the margin, provide tentative support for the impact of high magnitude transient stress perturbations across the ice sheet surface. Moreover, the interior migration of the equilibrium line with warming atmospheric conditions leads to an increase in bare ice area, potentially exposing existing crevasses over large areas that were formerly snow and firn covered. Finally, the inland expansion of lakes has the potential to fill crevasses, drive hydrofractures and deliver large volumes of water into subglacial environment and drive further transient fluctuations in ice flow and stress regime (Doyle et al., 2013). Future change of interior ice sheet flow is yet to be determined but observations indicate that the size and density of crevassing is increasing and our results suggest that such increases reduce albedo and thereby increase the proportion of absorbed downward energy available for surface melt.

### 4.4 Concentrated cryoconite

Increases in the fractional area of concentrated cryoconite are, contrary to expectation, not associated with a reduction of $\alpha_{MODIS}$ across the survey transect (Fig. 5C). This is, at least partially, explained by the fact that concentrated cryoconite only occupies a small fraction of the total surface area (1.6% maximum and 0.6% mean) (Fig. 6). The relatively small cryoconite footprint is explained by the nature of the cryoconite material. The filamentous, threadlike structure of cyanobacteria enables them to entangle debris and facilitate the formation of granules. These granules absorb greater solar radiation and melt into the ice forming cryoconite holes which melt down until in radiative and thermodynamic equilibrium and are typically just a few tens of centimetres in diameter (Takeuchi et al., 2001; MacDonell and Fitzsimons, 2008; Langford et al., 2010; Cook et al., 2015). Although this mechanism means that the hole has a very low albedo value when seen from directly above, the cryoconite itself occupies a small area and becomes hidden from non-zenith solar illumination resulting in an increased area-averaged albedo (Bøggild et al., 2010). Furthermore, some cryoconite holes form ice lids caused by the refreezing of water filling the hole during negative net radiation conditions, typically early morning when the sun is at its lowest elevation angle (Fig. 4F). While these frozen lids may undergo partial or complete complete midday ablation, their higher albedo presents a regulatory feedback that further acts to reduce the darkening effect of cryoconite holes.



A limitation of this study is that we underestimate the fractional area of cryoconite. Whilst cryoconite holes larger than 10 cm in diameter should be classified correctly, cryoconite holes smaller than 10 cm in diameter may not be. This is because holes smaller than 10 cm may not cover a large enough area in the corresponding pixel of the orthomosaic (15 cm width) for the pixel to be classified as cryoconite. Fountain et al. (2004) estimate that 23% of cryoconite holes in the McMurdo Dry Valleys

have a diameter smaller than 10 cm so this is potentially a large source of error. But if we assume that the cryoconite hole size distribution does not significantly vary across the survey transect, the underestimation of the fractional area of cryoconite should be constant. Therefore, there should be no change in the relative fractional area of cryoconite holes and our conclusion, that variations in the fractional area of cryoconite are not associated with $\alpha_{\mathrm{MODIS}}$ change, still stands.

Whilst the spatial pattern of $\alpha_{\mathrm{MODIS}}$ does not seem to be governed by cryoconite extent, temporal variations in albedo could

well be associated with change in cryoconite fractional area. Cryoconite has an extremely low albedo ($\alpha_{\mathrm{camera}}$ 0.10) and a spatial expansion of the extent of cryoconite will reduce $\alpha_{\mathrm{MODIS}}$. For example, Hodson et al. (2010) showed that 53% of plot-scale variation in albedo was correlated with growth of cryoconite holes and Chandler et al. (2015) report that the gradual seasonal reduction in albedo is correlated with an increase in cryoconite hole area. Possible causes for an increase in size and number of cryoconite holes are longer and warmer ablation seasons and increases in channelized meltwater which may promote sediment

supply (Cook et al., 2015).

## 4.5 Reconstructing albedo

The fractional area of each surface type was multiplied by $\alpha_{\mathrm{camera}}$ and summed to produce an alternative albedo product, $\alpha_{\mathrm{reconstructed}}$. Values for $\alpha_{\mathrm{reconstructed}}$ for the panels presented in Fig. 4 and 5 are listed in Table 3 and compare well with $\alpha_{\mathrm{MODIS}}$ with an RMSD of 2.4%.

$\alpha_{\mathrm{reconstructed}}$ presents an opportunity to improve the parameterization of bare ice albedo in surface mass balance models. Currently, these models either prescribe a background bare ice albedo (e.g., van Angelen et al., 2012) or simulate it as a function of the accumulated surface water (e.g., Zuo and Oerlemans, 1996; Lefebre et al., 2003; Alexander et al., 2014). For example, in MAR v3.2 ice albedo ranges between 0.45 and 0.55 as a function of melt rate (Alexander et al., 2014). However, significant discrepancies between the modelled and observed albedo are often reported because these schemes do not explicitly

consider the spatial and temporal variation of distributed impurities, crevassing and surface water in lakes and streams (e.g., Zuo and Oerlemans, 1996; Alexander et al., 2014), all of which have an impact on mesoscale albedo patterns.

The new scheme envisaged would use UAS or high resolution satellite imagery (e.g. WorldView) to classify the surface types in the bare ice area and attribute them to a background bare ice albedo field derived from MODIS imagery (e.g., van Angelen et al., 2012). The change in fractional area of surface types would then deterministically feedback into the bare ice

albedo field. For example, if the extent of ice algae were to increase, the fractional area of distributed impurities would increase which would act to reduce the bare ice albedo of the associated MODIS pixels. Likewise, higher summer temperatures would increase melt, causing the fractional area of surface water to increase and further reduce bare ice albedo. Currently, however, the temporal evolution of surface type extent in response to climate forcing is unclear. But if future work was directed at this problem, a more accurate parameterization of bare ice albedo in surface mass balance models may be achieved.





## 5   Conclusions

Using digital imagery and broadband albedo acquired by a fixed-wing UAS we classified and measured the albedo of six surface types that dominate the Greenland ablation area and its dark region. The fractional area and $\alpha_{\mathrm{camera}}$ of each surface type allowed us to investigate the relative contribution of each surface type to mesoscale ($10^2$ to $10^3$ m) albedo patterns, as
represented by $\alpha_{\mathrm{MODIS}}$. Our principle conclusions are as follows:

1) The primary control on $\alpha_{\mathrm{MODIS}}$ is the fractional area of uniformly distributed impurities. Although not the darkest surface type observed ($\alpha_{\mathrm{camera}}$ 0.27), distributed impurities dominate the $\alpha_{\mathrm{MODIS}}$ signal because of their extensive coverage across our survey transect.

2) Distributed impurities cover > 70% of the surface of the dark region and appear to be the cause of the regions low albedo.
In contrast, surface water (< 1.0%), concentrated cryoconite (< 0.5%) and crevasses (0.0%) have very low fractional areas in the dark region and do not appear to be responsible for the regions low albedo.

3) Where surface meltwater accumulates in sufficient quantities, such as in lakes or large braided streams, it has a significant impact on $\alpha_{\mathrm{MODIS}}$ due to its high absorbance properties ($\alpha_{\mathrm{camera}}$ between 0.19 and 0.26). But the 2% average surface water coverage across our survey transect can only explain 12% of $\alpha_{\mathrm{MODIS}}$ variation. We therefore conclude that the
accumulation of surface meltwater is a result rather than a cause of the darkening of the ablation area (Wientjes and Oerlemans, 2010).

4) Crevasses are only responsible for a small portion (17%) of $\alpha_{\mathrm{MODIS}}$ variation across our bare ice survey transect. However, we found examples that crevasses were able to reduce albedo by up to 10% in some locations, with wider and deeper crevasses causing higher reductions than smaller crevasses.

5) Concentrated cryoconite in holes and/or fluvial deposits only occupies a small fraction of the total surface area (0.6% mean) across our transect and was not responsible for reducing $\alpha_{\mathrm{MODIS}}$. The filamentous nature of cryoconite means that it facilitates the formation of granules which are subsequently hidden from non-zenith solar radiation. This process may actually increase mesoscale albedo in comparison to ice surfaces where cryoconite holes do not form.

6) As the area of bare ice increases in extent and remains for a longer period each year (Box et al., 2012), the surface mass
balance sensitivity to bare ice albedo will increase (Fettweis et al., 2011; Tedesco et al., 2016). Therefore it is important to improve the parameterization of bare ice albedo in surface mass balance models. Significant improvements may be made by representing the evolution of the surface types identified in this study. Currently, however, the relationships between surface type extent and climate forcing are unclear.



**\*Author affiliations**

Peter Sinclair (Earth Insight Foundation Inc., Wausau, WI, USA)

Gabriel Warren (unaffiliated)

*Acknowledgements.* J.C.R. is funded by an Aberystwyth University Doctoral Career Development Scholarship (DCDS). Fieldwork was
5   supported by Dark Snow Project crowd funding (http://www.darksnow.org/) and Villum Young Investigator Programme grant VKR 023121
awarded to M.S. A.H. gratefully acknowledges support from the Centre for Arctic Gas Hydrate, Environment and Climate, funded by the
Research Council of Norway through its Centres of Excellence (grant 223259). J.B. gratefully acknowledges support by a grant from the
Leonardo DiCaprio Foundation. AWS data were provided by the Programme for Monitoring of the Greenland Ice Sheet (PROMICE) and the
Greenland Analogue Project (GAP) via the Geological Survey of Denmark and Greenland (GEUS) portal at http://www.promice.dk.



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





| Surface type | Mean camera-derived albedo ($\alpha_{camera}$) | Standard deviation |
|---|---|---|
| Clean ice | 0.55 | 0.09 |
| Uniformly distributed impurities | 0.27 | 0.05 |
| Shallow water | 0.26 | 0.06 |
| Deep water | 0.19 | 0.04 |
| Crevasse and fractures | 0.11 | 0.02 |
| Concentrated cryoconite deposits | 0.09 | 0.02 |

**Table 1.** Mean $\alpha_{camera}$ and standard deviation (SD) for each surface type

| Date | $\alpha_{MODIS}$ vs. $\alpha_{pyra}$ | |
|---|---|---|
| | RMSD | $R^2$ |
| 6 August | 0.036 | 0.89 |
| 7 August | 0.033 | 0.95 |
| 8 August | 0.035 | 0.93 |

**Table 2.** RMSD and $R^2$ values for each albedo product for the three survey days



| Panel | Fractional area | $\alpha_{\text{reconstructed}}$ | $\alpha_{\text{MODIS}}$ |
|---|---|---|---|
| Figure 4A | Distributed impurities: 0.78 | 0.33 | 0.31 |
| | Clean ice: 0.22 | | |
| Figure 4B | Deep water: 0.021 | 0.32 | 0.28 |
| | Shallow water: 0.017 | | |
| | Distributed impurities: 0.78 | | |
| | Clean ice: 0.18 | | |
| | Cryoconite: 0.003 | | |
| Figure 4C | Deep water: 0.21 | 0.36 | 0.36 |
| | Shallow water: 0.13 | | |
| | Distributed impurities: 0.32 | | |
| | Clean ice: 0.34 | | |
| | Cryoconite: 0.006 | | |
| Figure 5A | Crevasses: 0.10 | 0.35 | 0.36 |
| | Distributed impurities: 0.54 | | |
| | Clean ice: 0.35 | | |
| | Cryoconite: 0.014 | | |
| Figure 5B | Distributed impurities: 0.50 | 0.41 | 0.44 |
| | Clean ice: 0.49 | | |
| | Cryoconite: 0.014 | | |
| Figure 5C | Distributed impurities: 0.61 | 0.37 | 0.35 |
| | Clean ice: 0.37 | | |
| | Cryoconite: 0.018 | | |

**Table 3.** The fractional area of each surface type for each panel presented in Fig. 4 and 5. Reconstructed albedo is equal to the sum of these fractional areas multiplied by the $\alpha_{\text{camera}}$ of each surface (Table 1).



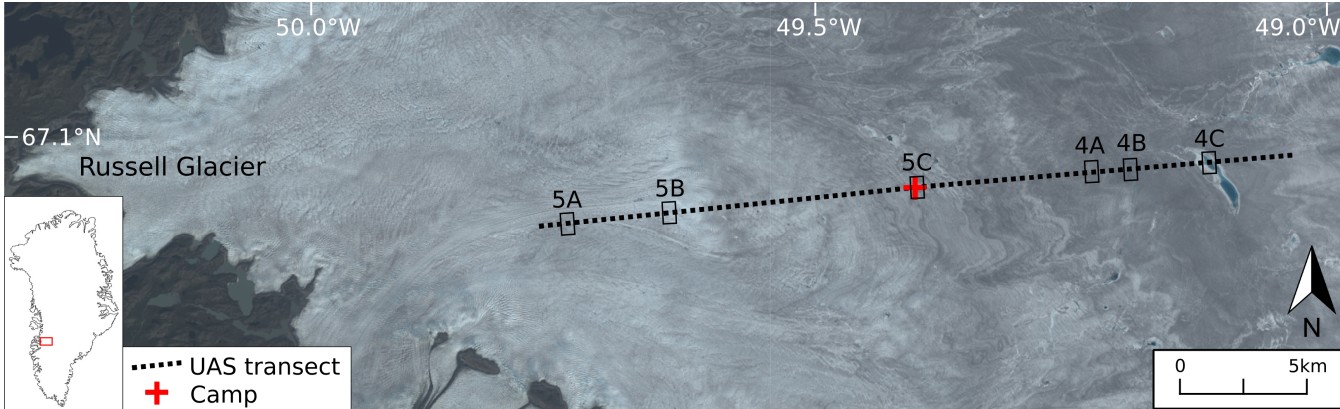

**Figure 1.** Landsat 8 Operational Land Imager (OLI) true colour image of the UAS survey transect across the Kangerlussuaq sector of the Greenland Ice Sheet on 6 August 2014. The letters designate the locations of panels in Fig. 4 and 5.

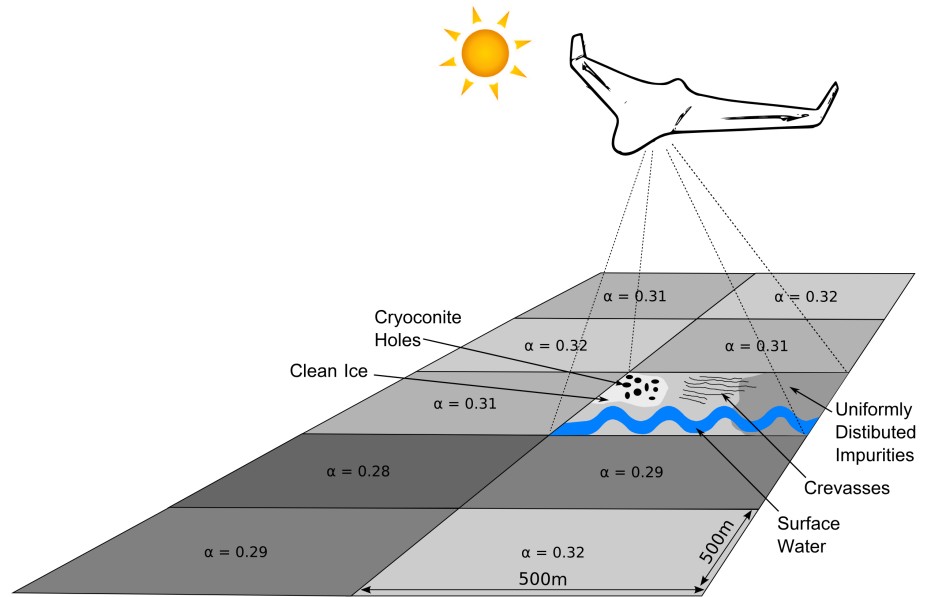

**Figure 2.** Schematic summarizing this study. Digital UAS imagery are used to classify the surface types that contribute to the albedo of a single 500 m MODIS pixel in the ablation area of the Greenland ice sheet.





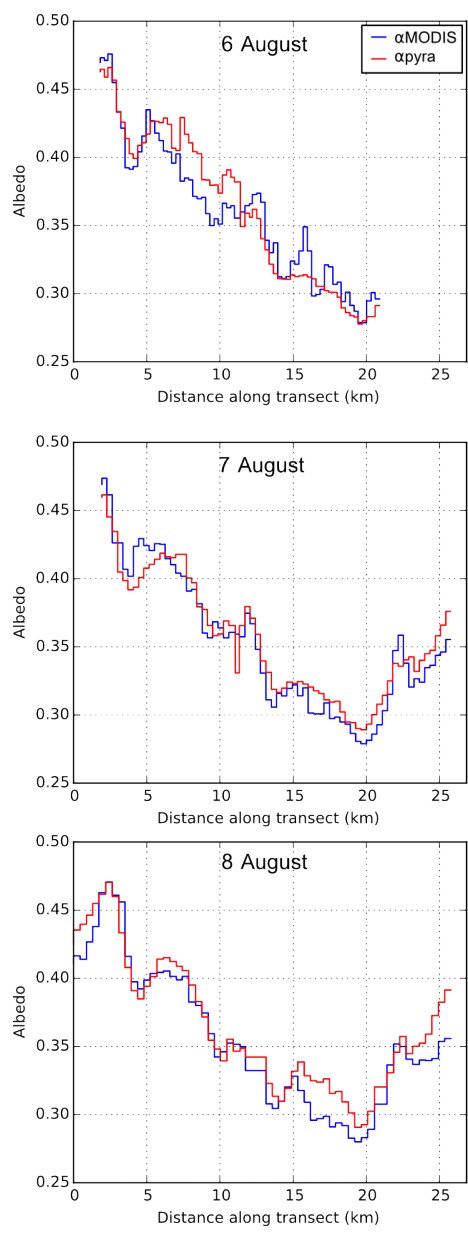

**Figure 3.** Plots showing the variation of MOD10A1 ($\alpha_{\text{MODIS}}$) and UAS SP-110 pyranometer ($\alpha_{\text{pyra}}$) across the survey transect. The RMSD and $R^2$ values are presented in Table 2. The location of the x-axis is shown in Fig. 1.



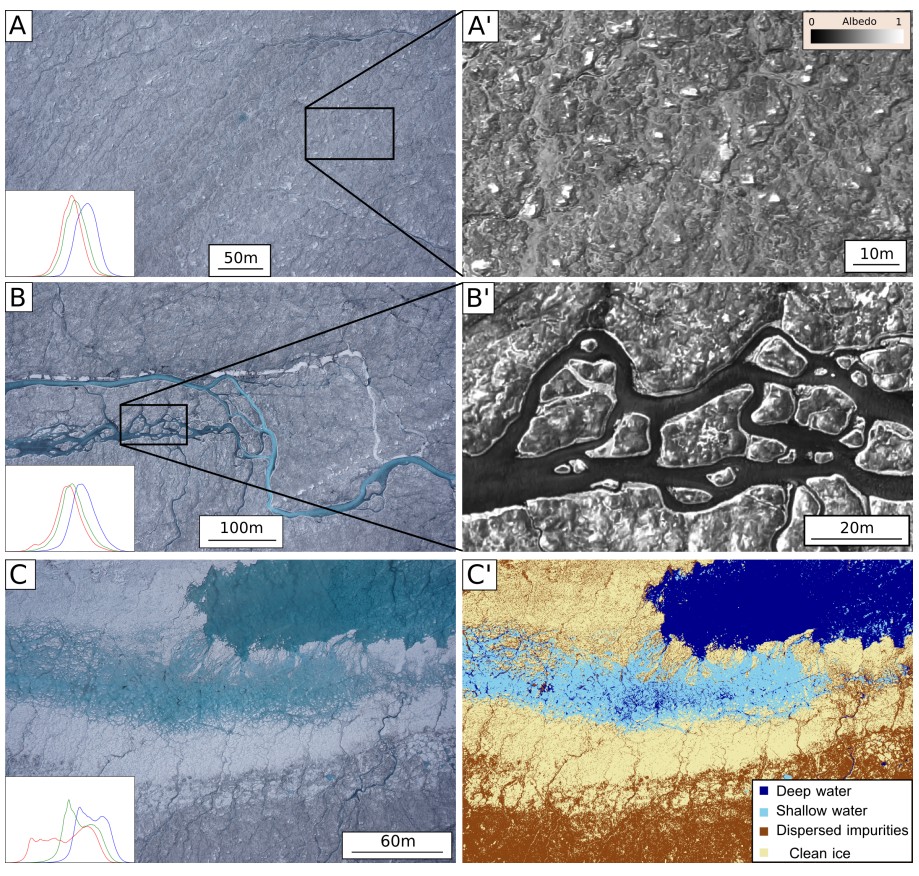

**Figure 4.** Representative surfaces across the survey transect, located in Fig. 1. (A) Uniformly distributed impurities with some clean ice caps. (A') The corresponding camera-derived albedo product ($\alpha_{\mathrm{camera}}$) (B) Channelized surface meltwater. (B') same as (A'). (C) Ponded surface meltwater. (C') Example of the classified surface types.



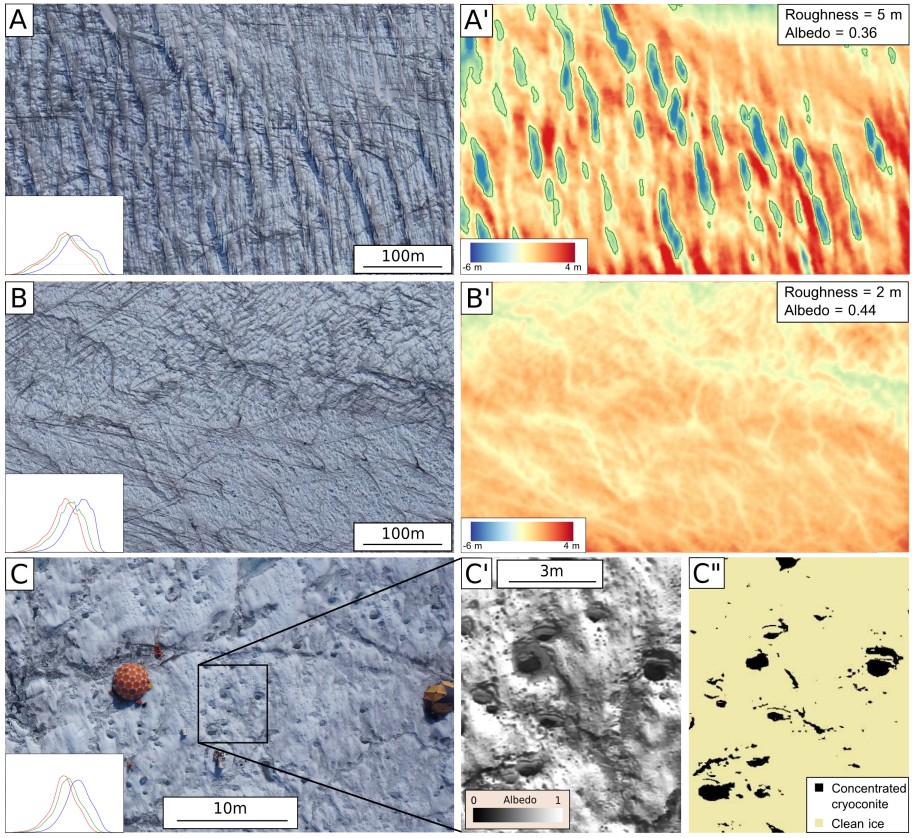

**Figure 5.** (A) Highly crevassed area. (A') Corresponding DEM with the crevasse troughs delineated. The area has average crevasse widths of 20 m and depths of 3 m and has an albedo 0.08 less than (B), an area with lower surface roughness and no crevasse troughs (B'). (C) Concentrated cryoconite in holes surrounded by clean ice. (C' and C") Example of classified surface types.





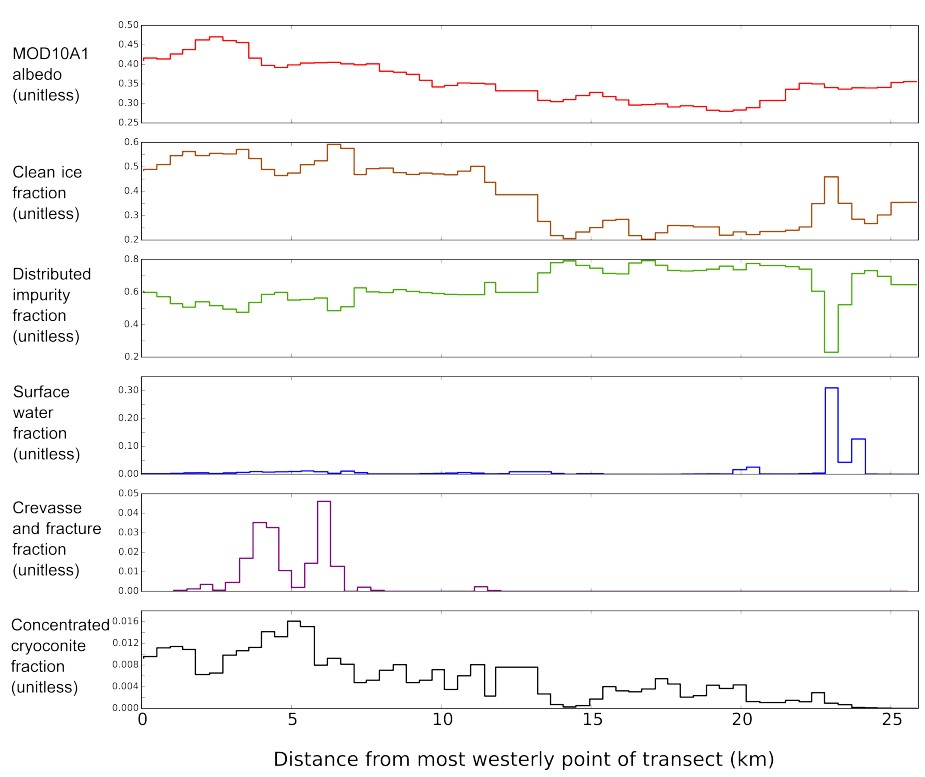

**Figure 6.** Variation of surface type fractional area across the survey transect. The location of the x-axis is shown in Fig. 1.