# Peer review of "Attribution of Greenland's ablating ice surfaces on ice sheet albedo using unmanned aerial systems"

_The Cryosphere, 2016_

## Referee Comment (RC1) · A. Pope (Referee) · 30 Sep 2016

Overall Comments: In "Attribution of Greenland's ablating ice surfaces on ice sheet albedo using unmanned aerial systems," Ryan et al. describe an interesting study where digital camera data, pyranometer measurements, and MODIS data are combined to better understand the range of surfaces present in one section of the Greenland Ice Sheet. Overall, the paper provides a new method for getting quantitative data using UAS as well as a thorough description of results, which will likely be of interest to the Greenland surface mass balance community. However, at times it feels like the paper over-reaches in significance without backing it up with a strong motivation. In addition, more detailed description is needed of the methods to be fully understood by

the reader. With some revision, this will be a solid contribution to The Cryosphere.

Broad Comments: Some overstatements: At a few points, namely in the introduction and the conclusion (and slightly in the title), it feels to me like the impact and applicability of this paper is overstated. The conclusions are very much limited to one section of the ablation region of the Greenland ice sheet – not the accumulation area and certainly not all of Greenland. Indeed, repeated mention of the "dark patch" reminds the reader of this (as that is confined to a section of West Greenland), but the text often discusses or draws conclusions to all of Greenland. These should be limited so as not to overstate the paper. In addition, the paper ties the spatial results here as being important to surface mass balance modeling. However, as the authors admit, nothing is known about the temporal changes of these fractions over time. As there are only 3 days of data, we cannot draw any conclusions from this paper either. Rather than providing such modeling applications as motivation/significance for this paper, I would suggest more simply stating that this paper gives some insight into higher resolution spatial variability and now temporal variability and trends need to be studied. I think being more direct will benefit the paper overall.

Describing Methods: Simply put, the methods need to be more fully described for the reader to be able to fully understand what the authors have done, particularly in the classification process. It is not clear what order things were done in, what was done manually vs. not manually, and while success is demonstrated with a few metrics, very few examples of other ways to assess the classification are given. While I very much appreciate a succinct paper, I do not believe there is quite enough for the reader to go on here. Specific suggestions would be a clearer description of classification seeding, processing, and validation, as well as more examples (figures – like 4C', but more of them!) showing where the classifier succeeds/fails. (On a related note – it is my understanding from k-means classifying of multispectral data that the classes are not necessarily distinct. A figure showing distinct clusters would very much boost my confidence in the results.)

Data & Code Sharing: "Copernicus Publications recommends depositing data that correspond to journal articles in reliable (public) data repositories, assigning digital object identifiers, and properly citing data sets as individual contributions." Currently this paper does not make it clear how data are shared, stored, or cited. It would benefit the paper and the community if both DATA AND CODE were shared following Copernicus recommendations and those of projects like "Geoscience Paper of the Future": http://www.scientificpaperofthefuture.org/gpf/node/1

Specific Comments: Title: "Attribution . . . on" doesn't quite make grammatical sense to me. Should there be an "effect" in there?

Title: I believe that it should be made clear this study is only done in one study area on the Russell Glacier / Kangerlussuaq area rather than implicating all of Greenland.

Authors: Is there really a need to call 2 more people the "Dark Snow Project Team" It seems unnecessarily complicated. Just add the two co-authors and add Dark Snow as affiliation if you like.

P1 L3: use of "10ˆ2 to 10ˆ3 m" comes up MANY times in the paper, but it seems like a confusing term to me. Do you mean to indicate a length scale or an area? If the former, then why not "100m – 1km"? If the latter, then it should be m2, and surely 10ˆ3 is not intuitive? Also, surely it is the combination of high spatial resolution of the imagery AND the scales that you are studying at that are important. Basically, use of this range has me a little lost. Please clarify / fix at many points in the paper.

Abstract: In general, sentence structure can be simplified to make this easier to read.

P1 L8: Suggest editing the sentence to "The highest correlation with mesoscale albedo was the fraction area of distributed impurities, which although not the darkest surface type, explains 65% of the albedo variability across the surface transect."

P1 L8: you haven't defined "distributed impurities" as a surface class yet.

P1 L14: Just my personal preference perhaps, but I do like the Oxford comma.

P1 L15: Example of what sounds like overreaching as I describe above.

P1 L20: There is high variability both before and after the snow disappears. See Ed Pope et al RSE 2016.

~P1 L20: The ablation region in Greenland is a relatively small area. Why is it so important to study it? Motivate this for itself and not just possibly later conclusions!

P2 L2: I have never heard the term "fracture cornices" – is it what it sounds like? Snow builds up in fractures more than the surrounding area and so persists?

P2 L6: To keep this paper up to date, it is important to note WorldView 3 (and WV4 which, depending on when you read this, will be launched soon or will have recently launched), which both have even higher spatial resolutions.

P2 L12: This range is essentially the same as satellites now. Which is fine, but that can't be the only motivation for UAS imagery.

P2 L15: No space after "lovenbreen". Oxford comma. Okay, I'll stop mentioning them now.

P2 L33: "approximate" rather than "consistent"?

P2 L33: I would also like to hear more here about the footprint size here, with a possible cross-reference to Section 2.3

P3 L1: What is the view angle/sensitivity of this pyranometer? How does that compare with the camera? How sure are you about this overlap?

P3 L6: Why 6 degrees?

P3 ~L25: What errors are there from flight issues, which aren't discussed here?

P4 L3: Change section title to "From Camera DN to Albedo" or something like that? Just trying to get away from calling the result "albedo" directly.

P4 L11: "EXIF" not defined.

P4 L14: Although you say you get another white reference panel every 10 minutes, lighting conditions can change MUCH faster than that. How is this accounted for? What errors might be expected?

P4 L16: This is the first of many examples, but where you provide a linear fit statistic, I think it is generally good practice to show us the fit. Things can have high r-squared value which not being truly linear!

P4 L12-22: This is a really important section but it is hard to read. Simplify sentence structure. Reduce the use of pronouns. Consider a flow chart.

P4 L25: And also a different bandwidth than the other sensors, too?

P4 L31&32: How was "resampling" done? Nearest neighbor?

P5 Section 2.6: I wrote this above, but I think it is crucial that you detail every step clearly and chronologically to make the classification process entirely clear. Description should include what is subjective/user-specific and what is not/repeatable. I would also like to see some figures which demonstrate the efficacy of your classification strategy.

P5 L16: For these segments, it would be helpful to see an illustration. Or are the bits in Figures 4 and 5 examples? Basically, I'm trying to figure out spatially how these are broken down. Maybe be more specific/explicit in Figures 1 and 2?

P5 Section 2.7: Rather than bilinearly interpolating MODIS (which is okay, but I'm not sure it is completely valid given that you are looking at smaller scale variability), have you tried to go the other direction? Do you have UAS imagery which completely covers the MODIS pixels?

P6 L5: Since you mention 7% uncertainty – have you thought about adding shading or something to the Figure to indicate the uncertainty?

P6 L7: Why only use August 8 data here? Could you consider "averaging" the data from the different transects to get a clearer signal? Just a thought – usually better to

use more data, right?

P6 L10: Figure 6 is out of order here.

P6 L14: I'm confused here. I though "water" was one distinct class (as listed earlier and in Fig 6), but here and in Fig 4 shallow and deep water are classified differently.

P6 L25: PCA should have a reference. Either something very basic or more applied to glacier facies (e.g. Pope & Rees 2014 RSE).

P6 L26: Why "r" here and "R2" everywhere else?

P6 L28: Why do you immediately equate PC3 and the presence of surface water? This gets re-written on P7 L28 and I'm sorry if I missed the motivation for this.

P8 L2: What about this sampling area in particular?

P8 L3: "absorption" instead of "absorbance properties"

P8 L5: Is there expected to be a daily variation in albedo? Does the daily product accurately capture this / compare to local noon best?

P8 L21: I understand that lakes are forming at higher elevations, but that isn't the subject of this paper. The ablation area is the subject here. And we don't have tons of evidence for consistent change in lakes in the ablation area (e.g. Pope 2016; Earth and Space Science), although there is certainly interannual and intraannual variability

P8 L32: Are the crevasses dry our water-filled? Does it matter?

P9 L15-20: The section seems a bit too speculative and disconnected from the ablation region. I think it needs to be a bit more tied to the subject of the paper.

P10 L1: Given the angles involved, do you think are missing small cryoconites with your imagery and also underestimating cryoconite that way? How does this matter in relation to MODIS?

P10 L4: The energy balance regime in the McMurdo Dry Valleys is completely different than the Greenland Ice Sheet. Would this not impact number/size/distribution of cryoconite?

Section 4.5 See earlier comments above.

P11 L3: You aren't studying the whole "Greenland ablation area and its dark region" – don't overreach!

P11 L18: "where" instead of "that"?

P11 Conclusion: I would try to condense this entire section if possible to be a little more concise. As written above, section 6 in particular is as simple as: "we have characterized the spatial, we don't know about the temporal variability. That temporal variability could be important for modeling." Don't overstretch.

Table 1: Have you considered making this a figure instead of a table? A box plot, for example? Or histograms? Provide evidence that you do have normal distributions if you're using std dev. Using a box plot would also show how distinct (or not) the populations are graphically. Annotate the figure if you want to explicitly provide the numbers, too.

Table 2. For the R2 values - if you're showing linear goodness of fit values, it would be good to show the scatterplot, too.

Figure 1: The colors printed out a little dark here. Replace "on" with "from"? "Fig." to "Figs."

Figure 2: I really liked some aspects of this figure and some of it confused me. I thought it was good to combine that you compare the MODIS and UAS imagery at different resolutions. What was less clear was the width of the UAS imagery and how that actually compares to MODIS pixels – and that you bilinearly interpolate MODIS data rather than averaging UAS data, which is what the figure makes It look like. I would also point out that MODIS is 500 m2; how does this compare with the "500 m2" chunks you describe in the paper? Should these actually be .25 km2 chunks? Just

checking. Also – there are 5 surface types listed here, but 6 in the paper.

Figure 3: I would suggest adding scatterplots (on the righthand aide) for each comparison to illustrate the linear relationship? See comments for Table 2.

Figure 4: Photos print a little dark. RGB histograms should be noted in caption. Give more examples of classifications like this! Consider making a 9-panel figure so each shows albedo and classification, for example?

Figure 5: Same goes here – show me the classifications to prove the do a good job!

Figure 6: Why are the lines different colors? It doesn't add anything to the figure. I would also suggest making a division of some kind (extra white space) between the top albedo part and the fractions. Consider labeling the two parts. When I first looked at it my immediate reaction was that the fractions added up to more than one . . . before I realized they weren't all fractions!

---

## Referee Comment (RC2) · Anonymous Referee #2 · 5 Oct 2016

This study represents a novel demonstration of aerial photography from unmanned vehicles to classify ice sheet surface types. The study goes on to explore the albedo statistics associated with six unique surface types, and includes an interesting discussion of some of the underlying mechanisms for albedo variability within each of these surface classes. The applicability of this analysis to Greenland-wide processes is limited because the study only explores albedo along a single transect over the course of 3 days. In one instance, I worry that the authors may have over-generalized a conclusion (described below), but in general I think the authors have maintained an appropriate scope and have not over-extended their analysis. The study is presented more as a demonstration of a novel technique which has the capability of leading to useful inferences. Clearly, quite a bit of data processing and analysis went into this study. Overall, I find this to be a very interesting and well-written study, and I recommend publication after minor revisions and comments below are addressed.

Major comments:

Conclusions #3 (p.11) states "We therefore conclude that the accumulation of surface meltwater is a result rather than a cause of the darkening of the ablation area (Wientjes and Oerlemans, 2010)" – Can this conclusion be robustly drawn from the rather limited spatial and temporal extent of measurements (25 km transect, only 3 days)? This conclusion might be legitimate for the domain that was studied, but I question whether it can be extended, based purely on the analysis presented here, to the entire ablation zone.

Minor comments:

Abstract and elsewhere: The term "mesoscale" is used here to represent scale lengths of 100-1000 m. In the meteorological community, mesoscale refers to scale lengths of several kilometers to hundreds of kilometers. Is there any precedence (in non-meteorological communities) for using "mesoscale" to describe sub-kilometer variability? If not, the authors may want to choose a different word to describe this scale.

p.2, 25: Consider changing "all-up", unless this is commonly used in this context.

P.3,7: It would be helpful to expand a bit on the "data cluster normalization" technique that was applied here. How exactly was this "compensation" achieved?

p.3,22: Does this RMSD refer to upwelling irradiance or to albedo? Please clarify.

p.3,31: "The relatively fast shutter speed minimizes image blur..." – True, although a ground speed of 25 m/s would imply movement of 2.5 cm while the shutter is open. This may be non-negligible compared with the stated pixel size of ∼11 cm.

p.4,23-25: "The uncertainty in alpha_camera is probably due to..." – While this statement seems valid, it would be more concrete to ascribe differences between alpha_camera and albedo determined from the CM3 pyranometer to differences in their detected spectral ranges (where the CM3 measures from 300-2800nm). Related to this, it might be helpful to show or describe the spectral response of the camera sensor that was used. Many digital cameras actually respond to near-IR light (see, for example, if your camera responds to a TV remote, as mine does), rather than exhibiting sharp cutoff at 700 nm as implied in the text.

p. 5,12: ". . . based on Euclidian distance to five equally weighted nearest neighbors." – Why five? This seems odd for analysis in pixel space, where I would expect 4 or 8 nearest neighbors.

p.5,25: ". . . based on a comparison between the GAP/PROMICE weather stations. . ." – for clarity (if I understand correctly), I would instead say ". . . based on a comparison with albedo measured at the GAP/PROMICE weather stations. . .". Furthermore, you might want to elaborate briefly (1 sentence) on how albedo was measured at these stations. Was it also with CM3 pyranometers?

p.6,5: "We found no detectable bias between alpha_MODIS over the three survey days." – Bias with respect to alpha_pyra? Or do you mean to say that there was no detectable difference in MODIS albedo between these three days? Please clarify the text.

p.6,14: "The fractional area of surface water (both deep and shallow) are well correlated." – Do you mean that the fractional areas of deep and shallow water are well correlated with each other? Again, please clarify.

p.6,18: Grammatical error.

p.9,8-11: Multiple grammatical errors.

p.9,33: "complete complete".

Figures 4 and 5: Please briefly describe the color histograms shown in the corners of

the figures and how they relate to the main figures.

---

## Referee Comment (RC3) · Anonymous Referee #3 · 7 Oct 2016

**Attribution of Greenland's ablating ice surfaces on ice sheet albedo using unmanned aerial systems**

Jonathan C. Ryan, Alun Hubbard, Marek Stibal, Jason E. Box, and the Dark Snow Project team

**Summary:**
The authors investigate the influence of difference surface types on Greenland ice sheet (GrIS) albedo within a region of exposed ice in the lower accumulation zone. They use an unmanned aerial system (UAS), which includes a set of pyranometers, a digital camera and other instruments. Measurements are taken across a transect spanning 25 km in the lower west coast ablation area of the GrIS for three days during the summer of 2014. Pyranometer measurements are compared with coincident measurements from the Moderate Resolution Imaging Spectroradiometer (MODIS). Fractional coverage of different surface types and their relative albedo is estimated using digital photographs. It is found that spatial variations in albedo at the ~500 m MODIS scale ("mesoscale") is primarily associated with spatial variations in the fractional area covered by distributed impurities. Streams, lakes, cryoconite holes are found to have a strong local impact, but do not contribute to this large scale variability. The authors suggest that future surface mass balance models for the GrIS should take into account the evolution of different surface types and account for their dynamic impact on surface albedo.

**General Comments:**
This study is an important contribution to our understanding of albedo in snow-free areas of the Greenland Ice Sheet ablation area. The work is highly important for improving simulations of surface mass balance in the area. The study is well thought-out, well organized and well written. It should be published after relatively minor revisions discussed below. A few general points:

(1) I think there needs to be a bit more discussion about the spatial scales of the processes and observations involved. While distributed impurities appear drive variations in albedo at the "mesoscale", variations in albedo at finer scales may be more sensitive to liquid water or cryoconite holes. At coarser scales, perhaps snow vs. ice coverage is more important. The authors should clarify that the results are valid at the "mesoscale", which is at the fine end of the spatial scale used by climate models and therefore important for that application.

(2) The conclusion that melting is a consequence rather than a cause of spatial variations in albedo is not really supported by the observations of this study, and overlooks the feedbacks linking melt to surface albedo change. See specific comments.

(3) Some additional details should be provided about differences between the measurements being compared, such as the spatial extent of different overlapping measurements, and the spectral range of different sensors. See specific comments.

**Specific Comments:**

1. **P. 1, Line 2:** Suggest changing "importance of distinct surface types on albedo" to "importance of distinct surface types on variations in albedo"

2. **P. 1, Line 18:** Change "ice melt (van den Broeke…" to "ice melt in ice-covered regions (van den Broeke…" to be a bit more clear.

3. **P. 2, Line 34:** What kinds of errors are introduced by variations in the height of the UAS as it flies? How these errors contribute to errors in the pyranometer and digital image albedo measurements?

4. **P. 3, Section 2.2:** Can the authors comment on the spatial extent of the area being measured by the pyranometers? While the pyranometers measure radiation over the entire hemisphere, there must be a radius within which surface albedo has a stronger impact on the measurements. This is important for comparison with MODIS because the 500 m MODIS pixel may not completely overlap with the area being "seen" by the pyranometers. This source of differences should also be discussed and mentioned as a source of error in the comparison.
   What does the 1 Hz sampling rate translate into in terms of the "spatial resolution" of the albedo measurements?

5. **P. 4, Line 11:** Please define and explain EXIF data.

6. **P. 4, Line 14:** Were the white reference measurements acquired by the UAS by taking a photograph of a reference target on the ground, or were the measurements taken on the ground using another sensor while the UAS was flying? Or was the reference target somehow flying on the UAS? Please clarify.

7. **P. 4, Lines 18-20:** This statement also needs clarification. Were the ground measurements taken at specific sites where coincident photographs were available, or were the photo-derived albedo of surface materials visible in the photographs compared against ground measurements of similar materials in other locations? How many photographs and measurements were used in this analysis?

8. **P. 4, Line 21:** Suggest changing "two products" to "the $\alpha_{camera}$ and surface albedo measurements" for clarity.

9. **P. 4, Lines 23-25:** The uncertainty results from the difference between the wavelength ranges of the two sensors; they represent albedo taken over different ranges. There is no defined wavelength range for albedo, although the 280 to 4000 nm range covers most of the radiation that comes from the sun. A "true" albedo would have an infinite wavelength range, but in practice a range must be defined. I suggest noting that the CM3 pyranometer covers a wider wavelength range, which covers the majority of the solar spectrum, and that the estimated uncertainty results from the shorter wavelength range of the CMOS sensor.

10. **P. 4, Lines 25-27:** 35% is quite a large percentage of the solar radiation. I would exclude mention of this as the important point is whether a shorter wavelength range can be used to estimate albedo over a longer wavelength range. Also the error from Corripio et al. (2004) is specific to that study. Corripio et al. (2004) estimate errors of 4-5% associated with a camera that does not capture NIR radiation using a radiative transfer model for snow, accounting for variability in NIR reflectance associated with different snow types. The error will be different for different materials. For the measurements of this particular study, the spectrum for ice, impurities, and water will be fairly uniform across all

wavelengths, and therefore the errors will actually be less than those estimated by Corripio et al. (2004). However, I think the authors should address and if possible, quantify this source of error more carefully, perhaps by examining the spectra of the different surface materials.

11. **P. 5, Line 4:** Was manual digitization only applied on a subset of the images? How many? Please clarify.

12. **P. 5, Line 21:** Please specify the wavelength range for the MOD10A1 product (300 to 3000 nm).

13. **P. 5, Line 24:** How is a "segment" determined? From Fig. 3 it seems that $\alpha_{pyra}$ is somehow averaged onto the 500 m MODIS grid, but this is not discussed. Clarify here, and perhaps also add details in Section 3.1.

14. **P. 6, Line 3:** Why not include $\alpha_{camera}$ in this plot? Since you are reconstructing the surface types from the camera measurements, showing that $\alpha_{camera}$ compares well to $\alpha_{pyra}$ and $\alpha_{MODIS}$ would help support the conclusions.

15. **P. 6, Line 5:** There was no detectable bias between $\alpha_{MODIS}$ and what other measurement? Do the authors mean to say that $\alpha_{MODIS}$ did not change detectably over the three-day period?

16. **P. 6, Line 9:** Figure 6 is discussed in section 3.2 but Figs. 4 and 5 are not mentioned until the discussion section. I suggest adding another paragraph here that qualitatively discusses Figures 4 and 5, introducing the different surface types. Figures 4 and 5 could also be moved to follow Fig. 6, but it may be useful to introduce the different surface types before discussing their fractional area and impact on albedo.

17. **P. 6, Line 24:** It seems appropriate to also cite Fig. 6 here, since it includes the fractional surface types.

18. **P. 6, Line 22:** Can a figure be added that shows the results of PCA analysis?

19. **P. 7, Lines 15-16:** Please explain in more detail how the RGB signatures are similar. Are the authors referring to the inset plots in Figs. 4 a, b and c and 5 a, b, and c? These insets should be discussed somewhere in the text. Also, since the authors do not have biological samples from these areas, it should be noted that this is not conclusive evidence that the RGB signatures are associated with algae.

20. **P. 7, Lines 24-25:** Other possibilities for the change in extent of the dark region should be noted, such as the consolidation of impurities due to increased melting.

21. **P. 7, Line 32 – P. 8 Line 2:** The statement seems to contradict itself in implying that increased fractional area of streams are not found in the dark region, yet there is a preponderance of meltwater there. Also, while meltwater may not directly drive spatial variability in albedo, melting is involved in the exposure, consolidation, and transport of impurities. One cannot conclude from the evidence presented that meltwater does not play an important role in the observed spatial variations in albedo. I suggest changing the last part of this sentence to note that the dark area cannot be explained as a direct consequence of spatial variations in fractional area covered by meltwater, and that the spatial variability at the MODIS scale is driven primarily by impurity concentrations, but noting that meltwater may play a role in the consolidation of impurities here.

22. **P. 8, Lines 22:** Suggest changing "will drive" to "will play an important role". Other factors such as further expansion of the dark region or more frequent exposure of bare ice will probably also play important roles as well.

23. **P. 9, Line 23:** Perhaps to be clear, note that a decrease in concentrated cryoconite is observed in the region with the lowest $\alpha_{MODIS}$ values.

24. **P. 9, Line 33:** Figure 4F does not exist. Please remove the reference or add a figure.

25. **P. 10, Lines 5-8:** Do you have any justification for this assumption? How do you know that the "distributed impurities" that are observed are not characterized by many small cryoconite holes?

26. **P. 10, Line 20:** It is not just the reconstructed albedo, but the combination of calculated albedos for individual components and measurement of their respective contribution to the reconstructed albedo that provides important information for improving models.

27. **P. 11, Line 6:** Since you have primarily looked into spatial variations, I suggest changing "primary control on $\alpha_{MODIS}$" to "primary control on spatial variations in $\alpha_{MODIS}$".

28. **P. 11, Lines 7-8:** The signal is dominated by the distributed impurities not just because of their extensive coverage, but also because of the large spatial variation in their fractional area on the "mesoscale".

29. **P. 11, Line 10:** Although as you note, the resolution of your measurements is not fine enough to capture ~25% of cryoconite holes.

30. **P. 11, Line 15:** Again, I don't think your results have proven that meltwater is a consequence rather than a cause of darkening. They do help support the fact that meltwater is not a *direct* cause of spatial variations in albedo in the area. Please revise.

31. **P. 11, Line 23:** The evidence for this is not really discussed in the paper. I think your main piece of evidence is the apparent increase in the fractional contribution of cryoconite in areas of higher albedo. Perhaps mention this here and earlier when discussing cryoconite.

32. **Table 2:** Change "each albedo product" in the caption to "$\alpha_{MODIS}$ vs. $\alpha_{pyra}$". Could $\alpha_{reconstructed}$ be included here as well? I think it would lend support to your conclusions about the usefulness of reconstructing, or deconstructing, albedo for improving climate models. (Table 3 does support that argument to some extent.)

33. **Figure 3:** I would suggest including $\alpha_{reconstructed}$ here as well as in Fig. 6. However, if you chose to only include these two sets of measurements, perhaps it would make more sense to include the $r^2$ and RMSD values on this figure instead of presenting them in Table 2. The fact that the transect is showing west on the left and east on the right should be mentioned in the caption or illustrated.

34. **Figure 4:** The small insets for Figs. 4a, b, and c do not have axis labels or a legend and are not mentioned in the caption. They are not mentioned specifically in the text either, although I think they show the RGB signatures of these images. Please mention in the text and include these details in the figure. Also revise "located in Fig. 1" in the caption to "shown in Fig. 1."

35. **Figure 5:** Again, please explain the small insets in Figs. 5 a, b, and c. Change "Highly crevassed area" to "Digital image of highly crevassed area". Also

explain that (B) and (C) are digital images. Explain that (B') is a DEM. Explain that (C') is a close-up showing albedo and (C'') is a closeup showing classified surface types.

36. **Figure 6:** Suggest showing $\alpha_{reconstructed}$ here as well.

**Technical Corrections:**

1. **Title:** "Attribution" does not seem to be the correct word here. Perhaps it should be changed to "influence" or "impact".
2. **Author list:** Why not simply include all authors in the authors list?
3. **P. 1, Line 8:** Change "The highest correlation with…" to "The property that exhibited the highest correlation with mesoscale albedo"
4. **P. 2, Line 12:** Change "generate" to "generated".
5. **P. 2, Line 15:** Change "Lovénbreen ," to "Lovénbreen,"
6. **P. 3, Line 6:** Change "were subsequently" to "was subsequently"
7. **P. 3, Line 13:** Change "SP-110 incorporate" to "SP-110 incorporates"
8. **P. 3, Line 14:** Change "and benefit" to "and benefits"
9. **P. 5, Line 24:** Change "have an RSMD" to "has an RMSD"
10. **P. 6, Line 14:** Change "fractional area" to "fractional areas", and change "($R^2$ 0.94)" to "($R^2 = 0.94$)".
11. **P. 6, Line 25:** Change "albedo patterns of albedo" to "albedo patterns".
12. **P. 7, Line 5:** Change "80% the dark" to "80% of the dark"
13. **P. 8, Line 4:** Change "is associated with" to "has"
14. **P. 10, Line 23:** Define MAR.
15. **P. 11, Line 24:** Change "($\alpha_{camera}$ 0.27)" to "($\alpha_{camera} = 0.27$)"
16. **P. 11, Line 9:** Change "regions" to "region's"
17. **P. 11, Line 11:** Change "regions" to "region's"
18. **P. 11, Line 13:** Change "$\alpha_{camera}$ between" to "$\alpha_{camera}$ ranges between"

---

## Referee Comment (RC4) · Anonymous Referee #4 · 25 Oct 2016

This study investigates Greenland albedo, a very important variable in understanding recent ice sheet melt increases and mass balance. Specifically, this study examines how fractional area of six different surface types and albedo vary along a 25 km long transect in Southwest Greenland. The distribution of six surface types were determined by analyzing high-resolution data collected with an UAV. The importance of each surface type in explaining albedo was quantified by correlating the first and second principle component of albedo derived from MODIS with the fractional area of each surface type. This investigation showed that spatial variability of distributed impurities was the most important surface types controlling albedo.

To my knowledge, this is the very first paper to ever publish UAV derived albedo and

surface type classification for the Greenland ice sheet. It is an important contribution to the literature about Greenland ice sheet albedo, and will most likely be followed by many other studies deploying UAV for this type of analysis. The text is easy to read. The discussion section is terrific and does a great job placing the findings in the context of existing literature. The presentation and analysis of this fascinating and rich dataset in the methods and result sections would benefit of a little bit of more work. Below follows some suggestion and comments for the authors to consider.

Main comments: —————

1. The ground resolution of the orthomosaics is reported to 15 cm. yet, many cryoconite holes are smaller than that, and the size of these holes may change of the season. This is mentioned in the discussion, but should be brought up in the introduction as well so that the reader knows from the very start that this study is limited to areas with large cryoconite holes. Consider calling it "area with concentration of large cryoconite holes". See more thoughts about this in minor comments below.

2. The principle component analysis should be described in the methods section. Among other thing mention which days of MODIS data were used to calculate PCs (August 8th?), where the PC rotated or unrotaed?, what is the overlap of the UAV footprint and the MODIS footprint, and so on. Finally, it would be interesting to see PC1 and 2 in a figure (e.g. Fig. 6)

3. I recommend a discussion about the uncertainty analysis of MODIS imagery. In particular, the discussion should bring up the differences in footprints between MODIS (pixel) and AWS data (point). In addition to the AWS point comparison, consider comparing the MODIS albedo with the average UAV albedo within each MODIS pixel. This comparison would be very interesting, as the UAV will capture the spatial variability within each pixel that the AWS station does not.

4. The finding that surface meltwater influence 12% of MODIS albedo despite only covering 2% of the surface is noteworthy. This points to surface meltwater having a

disproportionate influence on albedo relative to its area. Consider expanding on the analysis of the surface type influence on albedo by factoring in the areal extent of each surface type.

5. Reorganize the figures so that they appear in chronical order. In the manuscript Fig.6 is mentioned before Fig 4. and Fig 5.

6. Check the fractional area calculation of the surface types. In figure 6, clean ice covers ~50% of the area at the start of the transect, and distributed impurities covers ~60%. This adds up to more than 100%.

Minor comments: —————————-

Title: Confusing. . .rephrase. The sentence is difficult to understand.

L15: "Lovénbreen , Svalbard" » "Lovénbreen, Svalbard"

L7-8: Provide more details or references to the "data cluster normalization" method

L11: Add Apogee before SP-110 for consistency

L16: add equal sign in R2=0.85

L12-L22: The final steps are difficult to understand. Explain why the "linear least squares regression between the reflectance of surfaces within the illumination-corrected images and albedo of surfaces measured using the CM3 pyranometer from the ground" is needed. In the previous text it was explained that the CM3 pyranometer was used to establish the error in the SP-110 pyranometer, but here another reason seems to be given. Please clarify.

L3 − 8: Elaborate on the overlap of the MODIS and UAV footprint and the differences in spatial resolution. In figure 3 it looks like the aPYRA was resampled to MODIS pixels? If so, please mention this in the method section. Furthermore, a more comprehensive comparison of aPYRA and aMODIS would be extremely interesting in understanding MODIS sub-pixel variability (but perhaps beyond the scope of this paper).

L24: Rephrase this sentence. Clarify that 0.28 to 0.47 are albedo values. Furthermore, Figure 3 does not show that the variability in albedo is related to surface types. Please rewrite or include a measure of surface type area in Figure 3.

L26: Explain why this finding is "as would be expected"

L5-9. This text about how the cryoconite hole distribution might influence the conclusions is a bit problematic since the distribution of small cryoconite holes is unknown. However, it is not a big problem for this study. Simply state at the very beginning of the paper that your study is limited to features larger than 15cm. You can call it "areas with concentrations of large cryoconite holes" to clarify that smaller cryoconite holes exist but are not part of your study. Similarly, rephrase your conclusion and write "spatial patterns of aMODIS does not seem to be governed by the distribution of clustered large cryoconite holes" or something along those lines.

L18: Move the description of aCAMERA calculations to the methods sections.

Tables and Figures

Table 2: Mention the year in the table text

Table 3: Consider reformatting the table to make the fractional area items easier to read. For example list the various fractional area categories as sub columns to "fractional area", instead of a list for each figure.

Figure 4: Provide a legend for the histograms that are inset in panel A, B, and C. Also mention them in the figure caption

Figure 5. Explain the histograms (see comments to Fig. 4). Explain in the figure caption what the albedo mention in the box is about.

Figure 6: Clarify in the figure text if the data is from a specific day or based on averages.

---

## Author Comment (AC1) · 13 Dec 2016

We thank the reviewers for their thoughtful and constructive comments which will improve the manuscript. We attach our response to all the comments and provide an updated manuscript with the tracked changes.

The main changes include:

1) Moving the description of MODIS albedo to the start of the methods section. By doing this, the source of the medium-scale albedo patterns is defined early on and we no longer have to refer to a future section when describing the UAV camera and pyranometer albedo products.

2) Clarification and better validation of the classification method used.

3) Better description of albedo product validation (from both the digital camera and silicon pyranometers) using the CM3 thermopile pyranometers.

4) Focused the discussion on the ablation area of the ice sheet.

5) Included a new paragraph entitled "Spatial and temporal scales of albedo variability" at the end of the discussion which ties the discussion section together.

6) Cautioned reader that conclusions are only applicable to the 25 km survey transect and may not hold over the entire ablation area.